# Towards a Resilient and Resource-Efficient Local Food System Based on Industrial Symbiosis in Härnösand: A Swedish Case Study

Henrik Haller [1],*, Anna-Sara Fagerholm [2], Peter Carlsson [3], Wilhelm Skoglund [4], Paul van den Brink [1], Itai Danielski [1], Kristina Brink [2], Murat Mirata [5] and Oskar Englund [1]

1. Department of Ecotechnology and Sustainable Building Engineering, Mid Sweden University, 85230 Sundsvall, Sweden; paul.vandenbrink@miun.se (P.v.d.B.); itai.danielski@miun.se (I.D.); oskar.englund@miun.se (O.E.)
2. Department of Design, Mid Sweden University, 85230 Sundsvall, Sweden; anna-sara.fagerholm@miun.se (A.-S.F.); kristina.brink@miun.se (K.B.)
3. Department of Circular Economy, Chalmers Industrial Engineering, 41296 Göteborg, Sweden; peter.carlsson@chalmersindustriteknik.se
4. Department of Economics, Geography, Law and Tourism, Mid Sweden University, 85230 Sundsvall, Sweden; wilhelm.skoglund@miun.se
5. Environmental Technology and Management, Department of Management and Engineering, Linköping University, 58183 Linköping, Sweden; murat.mirata@liu.se
* Correspondence: Henrik Haller henrik.haller@miun.se

**Abstract:** The endeavour to align the goals of the Swedish food strategy with the national environmental quality objectives and the 17 global SDGs, presents an extraordinary challenge that calls for systemic innovation. Industrial symbiosis can potentially provide the means for increasing sustainable food production, using locally sub-exploited resources that can reduce the need for land, agrochemicals, transport and energy. This case study of the municipality of Härnösand, aims to assess opportunities and challenges for using waste flows and by-products for local food production, facilitated by industrial symbiosis. A potential symbiotic network was developed during three workshops with the main stakeholders in Härnösand. The potential of the COVID-19 pandemic to instigate policy changes, behavioural changes and formation of new alliances that may catalyse the transition towards food systems based on industrial symbiosis is discussed. The material flow inventory revealed that many underexploited resource flows were present in quantities that rendered them commercially interesting. Resources that can be used for innovative food production include, e.g., lignocellulosic residues, rock dust, and food processing waste. The internalised drive among local companies interested in industrial symbiosis and the emerging symbiotic relations, provide a fertile ground for the establishment of a local network that can process the sub-exploited material flows. Although there are multiple challenges for an industrial symbiosis network to form in Härnösand, this study shows that there is a significant potential to create added value from the region's many resources while at the same time making the food system more sustainable and resilient, by expanding industrial symbiosis practices.

**Keywords:** economic recovery policy; COVID-19; sustainable development; sustainability transitions; food supply chain; industrial symbiosis; circular economy

## 1. Introduction

The global food system currently constitutes the largest pressure exerted by humans on Earth, threatening not only local ecosystems but also the stability of the Earth system itself [1]. Currently, about half of the world's cultivable land area is used for agriculture [2]. As a growing and wealthier global population requires more biomass for food, energy, construction wood, and other biomaterials, the demand for land is expected to increase [3].

The economic cost of inefficiencies of the food sector surpasses USD one trillion a year globally, and up to USD two trillion if including social and environmental costs [4]. The food system has thus reached a point where systemic innovation and a radical transformation is necessary [5]. Since areas that can be sustainably used for food production may have already reached capacity [6], innovative methods of growing, sharing and consuming food are necessary to achieve an increased production within the planetary boundaries and without intruding upon other sustainable development goals (SDG). The transformation of food systems bears the potential to be a strong lever towards optimizing human and environmental health, and promote the transition towards sustainable development on local to global scales [1].

On a global level, an increasing number of policies are launched to trigger sustainability transitions. The European Green Deal, adopted by the European Parliament on 15 January 2020, is one such example that may have profound implications for the food sector. Within the EU, many member states have developed their own sustainability agendas for food. Sweden, for instance, has a national Food Strategy with an overall objective to ensure a competitive food supply chain, and to generate growth and employment. It aims to reduce vulnerability in the food supply chain and increase the level of self-sufficiency, while also achieving the national environmental quality objectives. During the Second World War and the Cold War, self-sufficiency and food contingency planning was a national priority in Sweden, but in the 1990s, as global markets opened up, domestic food production was assigned less importance [7,8]. The Swedish food system has moved from a high level of self-sufficiency to a structure dominated by specialized farms that are highly dependent on imported fuel, seeds, and machinery, as well as modern technology with sophisticated IT-systems and automated processes [9]. This development has increased the yield per hectare but has additionally made the food supply more vulnerable and dependent on global markets. In the county of Västernorrland for instance, there were slightly more than 7400 agricultural companies in 1970, and in 2013 the number was 2108; a decrease of 72%. The number of agricultural holdings throughout Sweden decreased by 57% during the same period [10].

The national Food Strategy noticeably aims to decrease the country's import dependency and, as a result of a defence bill from 2015, the Swedish Government sought to develop a new and modern "total defence", in which food supply is a central function [8]. A resilient supply of food, seed, tools, and inputs represents a crucial security concern for all nations, but how can this be achieved in a country like Sweden, where only 1.3% of the working population are farmers? Moreover, how can the food strategy be aligned with the national environmental quality objectives, the 17 global SDGs, and the national goal of becoming the world's first fossil-free welfare nation? The challenge is extraordinary, and new, innovative ways of growing, distributing and consuming food will be necessary to transform the food system accordingly. Strategies for resource efficiency such as industrial symbiosis (IS), where locally available wastes or by-products of one industrial activity become raw material for another, can potentially provide the means for increasing food production sustainably, using locally sub-exploited resources that can, directly or indirectly, reduce the need for land, agrochemicals, transport and energy. IS, being a systemic rather than a technological innovation, targets relatively deep leverage points and may thus have high potential for transformational change. It could create disruptive and deep systemic changes in several components of the food system, e.g., technologies, infrastructure, skills and knowledge [5,11–13]. In 2020, the Swedish Government adopted a national strategy for a circular economy that sets out the direction and ambition for a long-term sustainable transition of the Swedish society. The strategy promotes the use of IS as a means to achieve circularity. Circular economy applied to the food system implies, e.g., reducing the amount of waste, re-use of food, better utilization of by-products and food waste, nutrient recycling, and changes in diets toward more diverse and efficient food patterns [14]. Calculations undertaken by the Club of Rome indicate that a transition towards a circular economy in Sweden can result in over 100,000 new jobs, a 3% increase in GDP, and a 70% decrease in

GHG emissions [15]. In the coming years, the Swedish government will present policy instruments for companies and local governments to support circular business models and IS.

COVID-19 has unquestionably caused tragedy and poverty to many people, especially marginalized people in the most vulnerable areas. Nonetheless, there are many signs suggesting that the pandemic may also accelerate change and contribute to a transition towards a society that develops in sustainable, resilient, just, and equitable ways. The COVID-19 pandemic has induced the deepest economic downturn since the Second World War, requiring governments to design large-scale recovery plans [16]. While the post-World War II recovery generated an unpreceded worldwide economic expansion, the post-COVID-19 recovery may potentially act as the catalyst for an economy that operates within the biophysical limits of the planet [17], yet safeguarding the provision of a basic socio-economic foundation [18].

The awareness of the vulnerability of the socioeconomic system and the need for transition towards resilience have increased during the pandemic. A manifestation of this is an open letter, published on the 9th of April 2019 and signed by some of Europe's most important corporate CEOs, together with ministers and lawmakers. In the letter, they describe the recovery from the COVID-19 pandemic as an opportunity to revaluate our society and create a new model for wealth, based on green principles [19]. One week later, it led to the launch of the Green Recovery Alliance in the European Parliament. The implementation of that vision is yet to be realized on a larger scale, but a revaluation of the importance of resilient food systems has been observed in the media focus, as well as in the tendency among the public to stockpile food. Moreover, national governments have taken extraordinary measures to counteract the negative effects of the COVID-19 pandemic. The public funding spent to, e.g., save jobs and stimulate the economy, equates to many years of ordinary budgetary reforms [20] and many institutions have attempted unconventional and innovative methods to prepare for future crises. As a measure to recover from COVID-19 and reduce exposure to future shocks, policymakers in Amsterdam downscaled the global concept of doughnut economics [18] into a city model. Doughnut economics promote circular resource use to avoid overshooting the ecological ceilings (constituted by the planetary boundaries) but also includes a social foundation (based on the social aims of the SDGs) [21]. In April 2020, Amsterdam became the first city in the world to formally embrace this economic theory which has resulted in massive infrastructure projects and employment schemes [22]. Cities such as Brussels, Nanaimo, Dunedin, and Copenhagen decided to follow the example set by Amsterdam and are currently adopting similar measures. While doughnut economics delineate the ecological and social boundaries for a safe and just space within which food systems can operate sustainably, IS may provide operational guidelines that helps such a food system to materialise by promoting a more efficient, circular and just resource use.

The aim of this study is to assess opportunities and challenges for increasing the sustainability and resilience of local food systems based on industrial symbiosis, in Sweden, with the municipality of Härnösand as a case study.

The following objectives apply:

1. Identify major waste flows in Härnösand municipality and its surroundings, applicable to food production based on IS
2. Identify options for utilizing waste flows and by-products for local food production, and associated opportunities for building IS networks among local companies
3. Identify the main challenges and enabling conditions for widespread implementation of IS-based food systems in relation to (i) rules and regulations, (ii) consumers and markets, and (iii) knowledge and innovation
4. Discuss how behavioural and institutional reactions to the COVID-19 pandemic may be a catalyst for implementation of IS and the transition towards resilient and resource efficient food production systems.

## 2. Materials and Methods

In order to address the broad scope of this study, a consecutive chain of quantitative and qualitative methods was used. Figure 1 displays a methodology flow chart and the methods used are described below. Each subsection (Sections 2.1–2.3) corresponds to one research objective.

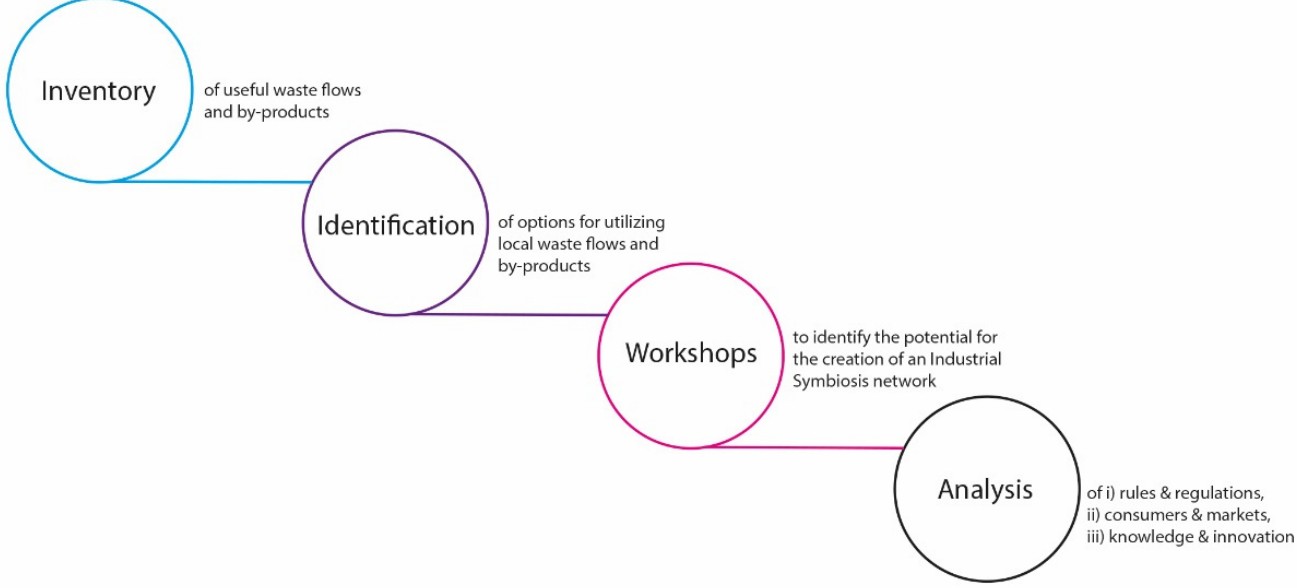

**Figure 1.** Flow chart of the methods included in the research strategy.

### 2.1. Inventory of Useful Waste Flows and By-Products

Four primary sectors; (i) forestry, (ii) agriculture, (iii) mining operations and (iv) food processing companies were selected and assessed on current flows of waste and by-products, to indicate the availability of different resources that could be employed within an innovative food system, facilitated by IS. Forest and agricultural areas were identified using recently compiled national land cover data with 10 m resolution [23] and aggregated to sub-watershed, municipal and county levels using geostatistical tools in GRASS GIS (GRASS Development Team, 2019). Forest volumes were identified using the SLU forest map [24]. Agricultural fields were identified from two national geospatial databases: agricultural "blocks", i.e., continuous agricultural areas often demarcated with physical structures, and "shifts", i.e., parts of blocks that are under specified management systems. Both databases were supplied by the Swedish Board of Agriculture. One block can thus be constituted of one or several shifts. Both the block database and the shift database are updated annually. For this study, data from year 2017 were used, containing over 1.2 million blocks and shifts, respectively. The block database contains (primarily) geographical location, block ID number, and type of land (cropland, two types of permanent cropland, pasture, wetland, other, and unknown). Further information about the blocks, such as areas under different management systems, can be found by searching in tabular data using the block ID. The shift data contains (primarily) geographical location and management system (crop type or land use). Areas under different management systems, as defined in the shift database, were calculated and aggregated to sub-watershed, municipal and county levels, using geostatistical tools in GRASS GIS. Extractable manure was estimated by first identifying the number of animals (cattle, pigs, laying hens, sheep/goats, turkeys, and other birds) registered in the national production unit database, administrated and provided by the Swedish Board of Agriculture. We then calculated extractable production of liquid and solid manure based on Einarsson et al. [25]. Two estimates were made, one for liquid manure and one for solid (liquid manure mixed with bedding material, e.g., straw). For cattle, the average per head value between "dairy cows" and "other cattle" was used. For sheep/goats and

turkeys, no estimation was made. For other birds, the per head value of broilers was used. Manure volumes were then aggregated to municipal and county levels, using geostatistical tools in GRASS GIS. An inventory of companies with by-products relevant for innovative food production was then created using data from the Swedish standard register [26] from the National Bureau of Statistics, complemented with information from grey literature. All companies registered in Härnösand were examined and those with activities related to food processing, forestry, animal husbandry, quarry mining, and fishery were included in the inventory.

*2.2. Options for Utilizing Local Waste Flows and By-Products for Food Production, and Associated Opportunities for Industrial Symbiosis*

A screening literature search was conducted to identify potential applications of residues and waste products in food production. The review included both scientific literature and grey literature, with the intention to identify a broad range of options with local relevance. In order to explore potential symbiotic connections between existing companies in Härnösand, a case-study method approach based on the *Guide for Industrial Symbiosis facilitators* [27] was used. Three workshops and two sets of informant interviews were organized during spring 2021 to collect relevant data. Chain sampling [28] was used to create the sample of respondents. This sampling technique was chosen as the population had had extensive previous contact with the research group. The selection was initiated by an interview with the development strategist at the Municipality of Härnösand (21 April 2021) to identify key persons in the private sector and in public institutions in the region. The three workshops lasted for 3 h, and the two informant interviews lasted for 2 h. The data collected, including texts and visualizations from workshops, were analysed by means of open coding in an iterative process with feedback between the data and the emerging themes of symbiosis. The first workshop (26 April 2021) focused on mapping local anchor companies in the symbiosis network, and developing a first version of a symbiosis map. The aim of the workshop was to create a solid foundation and to identify key collaborators for long-term symbiosis development. The second workshop (24 May 2021) was dedicated to creating commitment and engagement with participants by informing them about the potentials of industrial symbiosis. During the workshop, challenges and opportunities with symbiosis and what activities would be prioritized in the near future was discussed by the participating organizations. This was also the basis for an action plan for continued work. During the third workshop (21 June 2021), a more specific symbiosis map was developed using an iterative and participatory process based on the software Mural.

*2.3. Challenges and Opportunities for Widespread Implementation of Innovation in the Food System*

An assessment of challenges in terms of knowledge, innovation, institutional arrangements, and power was constructed based on information from grey literature and interviews with stakeholders from different parts of the food system (farmers, farmer's organizations and business incubators etc.). This assessment was structured according to the three strategic areas in the national food strategy: (i) rules and regulations, (ii) consumers and markets, and (iii) knowledge and innovation. The data was gathered during workshops with stakeholders engaged in the regional food strategy in the county of Västernorrland (1 and 26 October 2021). The sample of 20 respondents represented strategically selected stakeholders in the food system from the following organisations: Country Administration Board of Västernorrland, The Municipally of Härnösand, the regional incubator and science park Bizmaker, The Rural Economy and Agricultural Societies, High Coast Destination, Mid Sweden University, Research Institutes of Sweden (RISE), ICA local food store, Härnö Gin Distillery, The Federation of Swedish Farmers (LRF), and the consulting agency Macklean. During the workshop, a method with pick charts was used to analyse a number of action items. The pick chart method allows visual comparison of action items relative to their impact to the problem being addressed versus the ease/cost of implemen-

tation. This method is commonly used within organizations to sort and prioritize ideas for improvement [29].

## 3. Results

### 3.1. Major Flows of Waste and By-Products in Härnösand

The major material flows in Härnösand are generated by the agriculture and forestry sector but food processing companies and mining operations also generate by-products that could be used within an innovative food system, facilitated by IS. The results from the analysis of the land cover data show that the land use is dominated by production forest, both within the municipality and the county as a whole (i.e., Västernorrlands län). Only about 3% of the land is used for agriculture. The majority of locally sourced biological materials that could potentially be useful within the food production sector thus originate from forestry and associated industries (such as pulp and paper, timber, and wood-based products). Material flows from agriculture are considerably smaller, but since many materials originating from agriculture bear different characteristics than materials from forestry, and thereby can be used for other purposes, their potential contribution towards a resilient and resource-efficient food system should not be neglected.

Pine is the dominant tree species in the region of Västernorrland, closely followed by spruce (approx. 17% each). Other forests outside of wetlands are comprised of mixed conifer forest (10%), mixed conifer and broadleaf forest (14%), and broadleaf forest (5%). Temporarily non-forested (i.e., recently harvested) forests constitute 25% of the forest land. The majority of the forest volume is located in the western parts of the municipality but can be considered rather evenly distributed across the region. Annual forest productivity in the region is estimated to be 4.3 $m^3$/ha (cf. Swedish average: 6.1 $m^3$/ha) [30]. Tops and branches are left on ground or extracted and used for combustion in heating or combined heat and power (CHP) plants [31]. According to SCB [26], Härnösand has 606 registered forestry owners (code 02101) and 50 forestry companies (code 202200, 02109, 02102) [26]. Although production forest is the dominant land use within the municipality, there are no major wood industries, e.g., paper pulp plants or sawmills. The registered wood industries in Härnösand (code 16) include six planing mills, six building and carpentry companies (code 16231) and 16239), and ten companies in the group "other wood industries" (code 16292). Sawmills in adjacent municipalities could potentially be of importance for innovative food production, since they often generate large amounts of residues or by-products, typically used as wood fuel, for internal energy supply, and as raw material for paper pulp production [32]. Recycled paper and cardboard are also significant flows of fibre, derived from the forest industry. In 2017, the amount of recycled paper packages measured 565,700 tons [33]. Due to climatological conditions, the majority of agriculture in the municipality (68%) and region (65%) is comprised of ley and pasture. A notably large share of the agricultural land in the region (12%) is constituted of land that is classified as agricultural land but that is no longer used for agricultural production. In many cases, this land has been abandoned, resulting in bush encroachment. In other cases, the land is kept open for other purposes, such as sports activities or parking. The large share of "abandoned" agricultural land in the statistics highlights the documented decrease in agricultural production in the region. Moreover, it suggests that agricultural production could be substantially increased without conflicting with forest production or nature conservation.

There are 87 registered specialized agricultural companies in the municipality, including 51 crop cultivation (code 01110-01302), 21 beef cattle, and 15 milk producing (code 01410) companies. An additional 250 companies are registered as "mixed agriculture". Within cultivation, cultivation of "other annual and biennial plants" (code 01199) is dominating, with a total of 32 out of 51 companies [26]. There are also seven active stud farms and other types of horse breeders, and 12 active companies in sheep and goat breeding. One important resource from animal production is manure. Manure from cattle constitutes the vast majority of the extractable manure with a calculated annual produc-

tion of 2542 tons in the municipality of Härnösand, or 36,040 tons when including also adjacent municipalities.

As of 2020, four fishery companies are registered, one trawler, two sea water fishing companies, and one freshwater fishing company. Two commercial fish farms are also active [26]: the land-based Agtira and one sea-based located at Hemsön. Several food processing companies are established in the municipality, including one processing plant for meat, two fish processing companies, four bakeries and four other companies processing lemonade, coffee and tea, berries and vegetables [26]. There are also substantial nutrient-rich flows from wastewater treatment and slurry from biodigesters, with possible application in food production. Residual heat from the local energy company HEMAB and other industrial activities may additionally be used as heating for indoor food production.

The dominating bedrock in Härnösand is a type of sandstone called greywacke but diabase and amphibolite are also present. The latter typically contains more bioavailable nutrients, but both diabase and amphibolite have been shown to increase growth rates in trees better than lime [34]. An underexploited by-product, present in Härnösand, is rock dust from quarries that may be used for long-term nutrient supply in food production systems. Four active quarries exist in the municipality: two situated on greywacke, one on diabase and one on granite [35]. Investigations of the quality of the dust at these quarries will be needed to determine the applicability of local rock dust as a resource for local food production.

### 3.2. Options for Utilizing Waste Flows and Side-Streams for Local Food Production and Opportunities for Industrial Symbiosis Practices

The forestry sector has traditionally not been associated with food production but since the material flows from this sector in Härnösand are considerably greater than from any other land use sector, alternatives for lignocellulosic material have been highlighted. One promising application for forest wastes is production of growing substrates by composting. Currently, peat is the most commonly used ingredient in growing substrates, but other options are necessary since the use of peat is increasingly discouraged in many countries, due to sustainability concerns. Growing substrates are used for vegetable cultivation in greenhouses and within this niche, there are various sub-niches with high commercial values but more limited markets [36]. A thorough market analysis of the current and future needs for growing substrates in an emerging resilient food system in Härnösand is necessary to elucidate which market strategies are most appropriate. Another interesting possibility for utilizing forest residues in food production is rearing of edible insects. More than 2000 edible insect species have been identified and the majority of these naturally inhabit forests. However, most experiments to cultivate edible insects were conducted with feedstocks other than wood. Xylophagous (wood-eating) insects have a great potential since lignocellulosic residues are abundant and of low commercial value [37]. An ongoing research project in the region is investigation of the potential of rearing mealworms by using biosludge from the pulp industry as feed. Many waste flows can also be used for production of edible multicellular fungi, yeasts, single cell proteins, and fatty acids for human consumption or animal feed. Lignocellulosic waste is particularly suitable for white rot fungi, and agriculture and food processing waste may be used for algae cultivation [38]. Owing to high nutritional values, high feedstock conversion rates and low emissions of greenhouse gases [39,40], fungi, edible insects and single cell proteins are promising niches for local entrepreneurs with increasing markets. One of the important barriers against the development of these options was removed in Sweden in 2020 when the bans on commercialization of insects for human consumption were lifted. The major crop from the agriculture sector in Härnösand is grass, which is typically not associated with human food. However, grass can be used as raw material in biorefineries to produce high value products, e.g., protein concentrate for human consumption that can replace soy protein [41]. Other proposals put forward to make food systems more resilient and circular include recovering nutrients from manure and sewage sludge and cascading use of materials [14].

Opportunities for Industrial Symbiosis

Creation and diffusion of knowledge regarding business opportunities for underexploited waste flows play a vital role for industrial symbiosis networks to materialise [27]. The workshops with local stakeholders interested in IS contributed to both creation and diffusion of knowledge about techno-physical possibilities, and resulted in increased awareness regarding potential symbiotic exchanges of waste material, water and energy between existing companies, as well as niches to be filled by start-ups (Figure 2). Moreover, these provided guidance on areas where further knowledge is needed (market conditions and institutional conditions, which we thereby explored further).

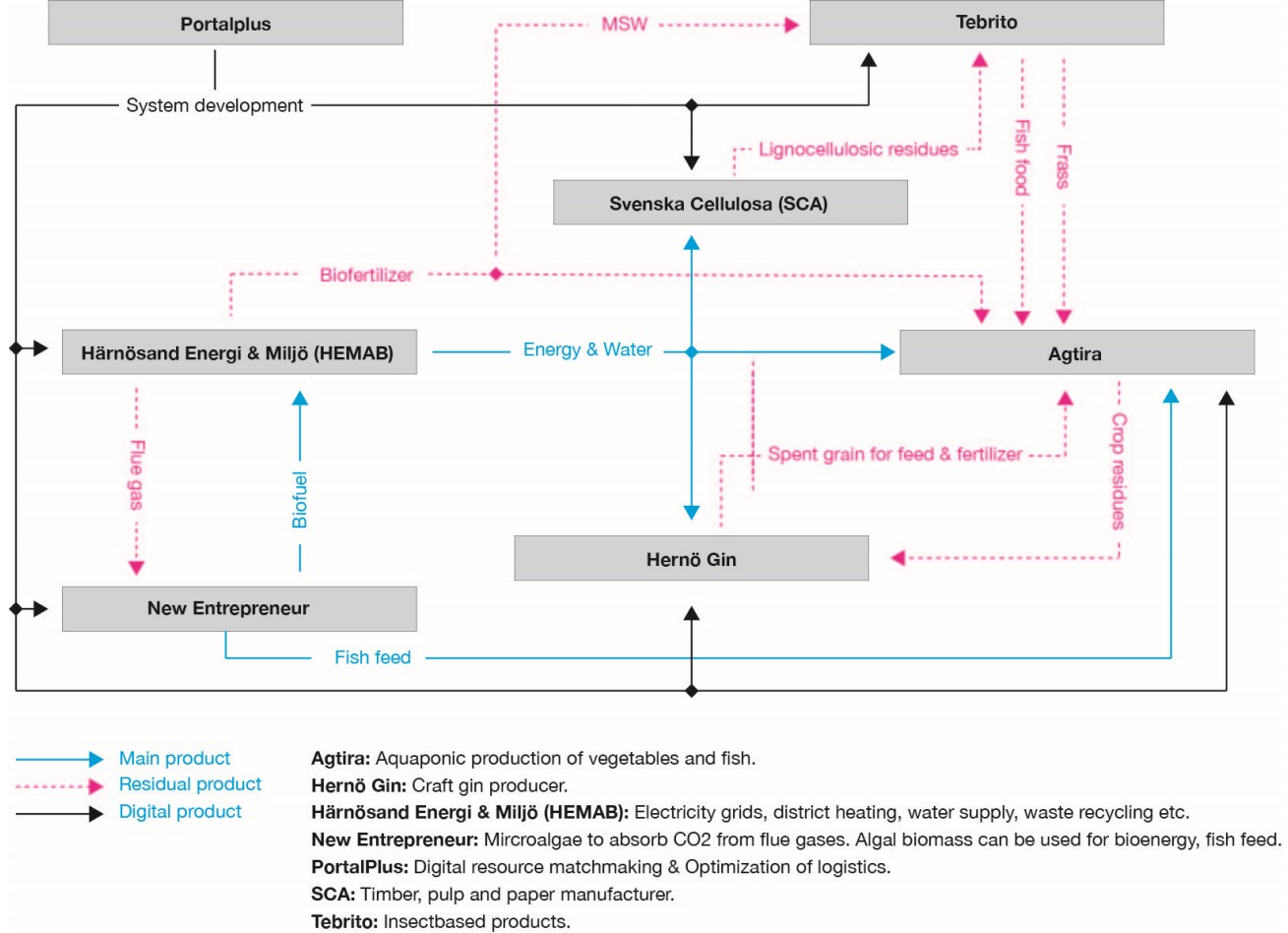

**Figure 2.** A proposed industrial symbiosis network in Härnösand that emerged during the workshops.

A potential symbiotic network was envisaged around the main stakeholders who are interested in, or currently work with, IS. The municipality-owned company Härnösand Energi Miljö AB (HEMAB), a key actor in the network, is involved, e.g., in district heating, waste recycling, water and sewage services, electricity grids, wind power, and biogas production. HEMAB can provide fertilizer from compost and biosludge to the vegetable cultivation systems at Agtira. MSW from HEMAB can also be used to feed insects, reared at Tebrito. Tebrito is currently operating in Orsa, in the county of Dalarna, but if their experiments with biosludge from the pulp waste from SCA emerge successful, they may intend to open a branch in the region. Härnösand would provide an interesting location owing to the emerging IS network and proximity to the source of the biosludge. SCA has a pellet fuel production plant in close proximity to the other companies, which can provide heating to Agtira's greenhouses and organic waste as feedstock for insects, fungi or single

cell proteins. As a CCU (carbon capture and utilisation) application, the $CO_2$ from the flue gases at HEMABs district heating plant may be used for microalgae farming. Algal biomass from such new entrepreneurs can provide HEMAB with biofuel or fish feed for Agtira. Agtira in turn, receives energy and water from HEMAB and may receive fish feed from a number of sources, including spent grain from Härnö Gin and insects from Tebrito. The insects' excreta (frass) from Tebrito may be used as fertilizer and biostimulant for the vegetable cultivation at Agtira. The craft gin producer Hernö Gin receives energy and water from HEMAB and may use rejected or damaged crops from Agtira as a source of fermentable carbohydrates in the production of ethanol. The IT company PortalPlus may facilitate a digital matchmaking service and transportation logistics of the resources to be exchanged. Challenges and opportunities for such an IS network to arise, in terms of rules and regulations, consumers and markets, knowledge and innovation, are addressed in Section 3.3. IS presents a complex endeavour and organisational support is typically needed to engage stakeholders in a coordinated effort among multiple private and public stakeholders. A financially secured facilitation organisation, such as the municipality is thus necessary for IS networks that do not emerge spontaneously [5,27]. An incremental innovation approach, in which local food entrepreneurs are supported by researchers to develop new production methods, may catalyse the development of an IS network in Härnösand.

### 3.3. Challenges and Enabling Conditions for Widespread Implementation of an Innovative Food System in Härnösand

Respondents from different segments of the food system in Härnösand highlighted a number of challenges and enabling conditions that are crucial for accomplishing a resilient food system. The respondents' input, arranged according to the three strategic areas in the national food strategy, is presented below and summarized in Table 1.

**Table 1.** Overview of challenges and enabling conditions for widespread implementation of an innovative food system in Härnösand.

| Strategic Area | Challenges | Enabling Conditions |
|---|---|---|
| Rules and regulations | • perceived technical and legislative barriers to increase resource efficiency and resilience <br> • need for adaptation and simplified application of regulations <br> • regulations make it difficult to redistribute unclaimed food <br> • the most suitable land for establishments of an IS network is currently not owned by the municipality | • National food strategy support easier implementation for structural changes in legal structures <br> • the elimination of the ban on commercialization of insects for human consumption <br> • National strategy documents such as the national strategy on circular economy can be used as a lever to attract funders and catalyse implementation of food system innovations <br> • scale-specific policies adapted for conditions in Härnösand |
| Consumers and markets | • need to emphasize the "value of the origin" of local products on the regional, national and international market | • local production is equally attractive to consumers as organic production <br> • consumer knowledge and innovative food environments can catalyse change towards sustainability |

**Table 1.** *Cont.*

| Strategic Area | Challenges | Enabling Conditions |
|---|---|---|
| Knowledge and innovation | • need for technical assistance, knowledge, and research on sustainable and innovative food production<br>• limited investments in research and development<br>• limited cooperation and knowledge exchange between academia and food industry<br>• large, sparsely populated areas and long distances between facilities,<br>• aging population, net migration away from the county<br>• lack of processing industries in the region | • many potential resources, such as residual heat, treated wastewater, slurry from biodigesters, residues from fisheries, and wastes from the paper and pulp industry are currently subexploited<br>• increasing number of innovation-driven companies have emerged in Härnösand in recent decades |

### 3.3.1. Rules and Regulations

The national food strategy in Sweden emphasizes that it should be easier to implement structural changes in the agricultural and horticultural industries, and businesses should be able to operate through efficient legal structures irrespective of size. Rules and regulations in an individual industry or sector affect the capacity of that specific sector to develop, and has an impact on the conditions for starting, running and developing businesses in the food supply chain. In Härnösand, food production is affected by several policy objectives, and the respondents perceive the presence of a number of technical and legislative barriers to shifting towards higher resource efficiency and resilience. The respondents highlighted a need for better dialogue between authorities, adaptation and simplified application, interpretation, and follow-up of regulations. Current regulations make it difficult to redistribute unclaimed food and the respondents called for new incentives to reduce food waste and surplus, such as food quotas and standards [14]. The land ownership may present a limiting factor for the establishment of large scale IS networks. Although many companies have expressed interest in adhering to IS networks and the municipality is eager to support this initiative, the most suitable land for new establishments is currently not owned by the municipality. Policy and legislative changes, such as the elimination of the ban on commercialization of insects for human consumption in 2020, may represent a window of opportunity for entrepreneurs. Exploiting national strategy documents such as the national strategy on circular economy, or legislation about city development or infrastructure plans that include a focus on sustainability, can be used as a lever to attract funders and catalyse implementation of food system innovations [27].

### 3.3.2. Consumers and Markets

Challenges related to consumers and markets include consumer habits, such as reducing food waste throughout the food system and increasing the degree of self-sufficiency. One item that many respondents highlighted is the need to emphasize the "value of the origin" of their products on the regional, national and international market. Consumer demand is a key component in the transformation to a sustainable food industry, and respondents stated that local production was at least as sought-after as organic production. Bearing in mind that a majority of the food in the region is imported, there is a significant potential for increased local production. Consumer knowledge and innovative food environments are important catalysts for change towards sustainability. On this topic, Grunert [42] has elevated the emergence of more demanding and critical food consumers, and Isenhour [43] describes how the growing consumer interest for sustainable food con-

sumption can be used as a lever. Campaigns that seek to promote local, resource-efficient food production may thus be used to create an increased demand for an innovative local food system.

### 3.3.3. Knowledge and Innovation

Most food systems in Sweden still build upon a linear use of resources and the lack of circular business models for food production which means that many potential resources, such as residual heat, treated wastewater, slurry from biodigesters, residues from fisheries, and wastes from the paper and pulp industry, are left unexploited [44,45]. The respondents listed technical assistance, knowledge, and research on sustainable and innovative food production as prioritized areas. Challenges in Härnösand related to objective knowledge and innovation are, largely, similar to national challenges regarding a sustainable development of the food industry. For example, the Swedish food sector is characterized by small and medium-sized enterprises and there are only a few larger companies conducting industrial research. There are also limited investments in, and low focus on, research and development, a low degree of cooperation amongst actors within the sector, and limited cooperation and knowledge exchange between academia and industry [46]. The regional characteristics, such as large sparsely populated areas and long distances between facilities, and the sociocultural heritage, pose region-specific limitations as well as opportunities. In the county of Västernorrland, the standard of living is relatively high and there is potential for growth, but, at the same time, the region faces major challenges with an aging population, low levels of higher education, and net migration away from the county due to urbanization. A noteworthy dynamic with potential implications on the innovation capabilities is linked to challenges in attracting younger people to the food sector. This is reflected by the number of farmers older than 60 years increasing from 33% to 48% between 1996 and 2013, and those under the age of 35 decreasing from 7% to 5% during the same period. Moreover, there is a lack of processing industries in the region, which limits the interest in innovation and further development of food production systems [10]. However, the establishments of innovation-driven companies that have emerged in Härnösand in recent decades indicates an interest in, and capacity for, substantial innovation. An example of this is the innovative, multifunctional production systems for tomato and fish that is the largest commercial aquaponics system in Europe [47]. There are also important institutional developments with good potential to foster innovation capabilities in the region. For example, when Mid Sweden University closed its campus in Härnösand in 2013, the municipality created a Growth Department, focused on innovative food production, and the municipality aspires to be the hub of development and innovation for circular food production in Northern Sweden. In collaboration with Mid Sweden University, the municipality has invested public finances to support innovation within the food sector.

According to the Swedish Bureau of Statistics, there are, in total, 360 companies connected to agriculture in Härnösand (including family-owned farmers). Such companies, and their employees, within traditional food production represent an important resource of competence and knowledge, in addition to being providers of arable land and biomass resources. The presence of an innovative biorefinery with knowledge and experience (*Domsjöfabrikerna*) in an adjacent municipality is another valuable resource for developing resilient food systems. Other knowledge resources for development within the food sector include *Eldrimner*, the national resource centre for artisan food production. *Eldrimner* is located in the neighbouring county of Jämtland/Härjedalen about 220 km from Härnösand. It has been described as bearing an important role in adapting cultural heritage and enabling entrepreneurial possibilities in the growing artisan food production sector [48].

### 3.4. The COVID-19 Pandemic as a Catalyst for the Transition towards a Resilient Food System

The COVID-19 outbreak has accelerated the consciousness and underlined the fragility of our societies, in terms of, e.g., food and water supply, health, and economy [49]. The COVID-19 pandemic may thus disrupt some patterns of production, distribution and

consumption of current systems and accelerate institutional adaptations towards a resilient food system, since social systems typically respond to external or internal pressure through reorganisation, learning and adaptation [13,50]. Although the long-term effect on policy and economy are yet to be seen, the crisis has the potential to act as a focusing event [51] and may affect policy change by directing public attention and financing to a number of sustainability concerns. Multiple institutional adaptations and smaller system changes that may take the local food system in Härnösand towards resilience and higher resource efficiency were seen during the pandemic. A project named *Industrial Symbiosis for competitive food production in Härnösand*, approved in 2020 and financed by the municipality of Härnösand and Mid Sweden University, is a direct result of the awareness raised by the COVID-19 pandemic, considering the vulnerability of the current food system and the need for restructuring. As a result of an appeal published in Swedish newspapers on the 20th of March 2020 [52], by small-scale and artisanal food producers, a number of retailers, including ICA, Axfood, COOP and LIDL, decided to change their procurement policies to favour locally produced food. According to a survey in which more than 300 artisanal food producers participated, 82% of the respondents had experienced an unchanged or increased demand for artisanal food products during the COVID-19 pandemic, and 59% believed that the COVID-19 pandemic would benefit their business [53]. The following local policy change in Härnösand, sparked by the pandemic, shows that a crisis may initiate new strategic alliances. During the pandemic, the food unit of the municipality initiated a collaboration with grocery stores concerning food waste. The food that would otherwise have been discarded in stores, due to passed expiry dates, is now collected by the municipality, screened for quality, and supplied to children in schools and preschools and for the elderly. The global home-gardening boom that seems to have been fuelled by the fear of food insecurity and the fact that many people have spent more time at home [54] is another short-term effect of the pandemic that has also influenced the politics. As an effect of higher tax revenues than expected for 2020, the municipal board of Härnösand decided to bring forward the launch of a program devoted to reducing unemployment by promotion of urban farming projects.

## 4. Discussion

The food system in Härnösand is driven by intricately connected economic, social, cultural, and ecological factors. The linear resource use and high dependence on mined raw materials associated with the current food system in Härnösand are incompatible with national and international sustainability goals. A more efficient, circular, resource use promoted by IS may prevent Härnösand overshooting its pressure on Earth's life-supporting systems and the promotion of social symbiosis may prevent vulnerable groups falling short on life's essentials (nutrition, housing, education, healthcare, political voice etc.). The shift towards sustainability and reduced vulnerability in the food system in Härnösand requires transformations on many levels, including new business models and innovative production and distribution methods. The disruption in global supply chains due to COVID-19 has borne numerous short-term undesired effects, but it has also accelerated the awareness regarding the importance of resilience in terms of food and water supply, health, and economy. This increased awareness seems to have instigated some institutional adaptations in Härnösand that may, in the long term, translate into transformative actions towards a resilient food system. In complex systems such as the food system, tensions may arise between resilience and resource efficiency [55,56]. A resilient food system i.e., a system with the capacity to maintain and protect provision of sufficient, appropriate and accessible food to all, in the face of various and even unforeseen disturbances over time [57,58] may thus depend on certain redundancy (duplication of critical systems functions with the intention of increasing reliability). Trade-offs, between redundancy to safeguard important functions on one hand, and resource efficiency on the other, may have to be employed for an innovative food system based on IS. However, Parker and Svantemark show a number of case studies where IS has contributed to resilience as

well as resource efficiency in food production [11]. Well-designed local networks based on IS may thus be both resilient, since they constitute a united network that is stronger together than the sum of its individual parts [27] and resource efficient, since waste flows are exploited. A high reliance on local resources may increase the resilience of food systems [11] and the identification of locally available material and energy flows in Härnösand that can be used in this process thus remains essential.

Although the material flow inventory in this study only partly quantified the volumes of different waste flows, it revealed that many underexploited resource flows are present in quantities that render them commercially interesting. The upcycling of such low-value by-products into high-value food products can not only support the local food system with feedstock, but also create added value in the region and make material flows more circular. The identified resources from land-use and organic by-products from the food processing industry may be used in numerous innovative ways, as exemplified here. The inclusion of such by-products may increase the resource efficiency significantly but any such strategy must also safeguard that the peoples' needs are met without overshooting Earth's ecological ceiling in order to be considered sustainable. The commercial value, market segments, size, and potential buyers for the material flows vary widely. Some production systems are already sufficiently mature to be commercialized, others are in need for further research. Many key factors are in place in Härnösand that may facilitate exploitation of waste flows and by-products and adoption of IS-based food production. Much of the existent infrastructure can be refurbished for IS. The internalised drive among local companies interested in IS that can act as vocal supporters and the emerging symbiotic relations between a few of them, together with a local government devoted to support innovative food systems, provide fertile ground for the establishment of a local IS network that can process the sub-exploited material flows. The promotion of innovative methods to distribute food, and alternative business models such as community supported agriculture and farmers, may be used to shorten food supply chains [14]. While hampered by a fragmented organizational structure, the food production sector in Härnösand has access to knowledge resources as well as a culture for innovation that could drive development towards implementing innovative and circular business models for increased food production. The growing public awareness about IS from positive experiences may create pressure from consumers, employees, and investors to promote innovations based on IS [5]. Experiences from other regions show that the existence of a steering group or an incubator, whose role is to facilitate exchanges of knowledge, and resources between actors with a common goal of resource efficiency and resilience, is crucial [59]. The municipality may play a particularly important role in coordinating and conducting symbiosis development activities between different actors involved in the food system [60]. The long distances between facilities, and the fact that the most suitable land for new developments within a confined IS network is currently not owned by the municipality, are limiting factors that need to be addressed. The geo-economic challenges in Härnösand could be partly met with the help of centralized digital platforms, for transferring information between stakeholders in the region. Such platforms should be widely accessible, transparent, and reliable. The core of such digital platforms is databases that could include a wide range of information, including company information (e.g., knowledge, expertise and products) and associated environmental (e.g., resource use, by-products, wastes, emissions, and animal welfare) and social (e.g., working conditions) aspects. Information could also be collected about consumer satisfaction and product reviews. Creating digital channels with access to such information could stimulate the circular economy and contribute to formation of IS symbiosis, i.e., by matching by-products from specific stakeholders to the needs of others. Databases could be used as input to dynamic system models, which could be used to evaluate, optimize, and identify scenarios for improved material and energy flows. Consumers could access database information via a graphical user interface and learn about the different aspects of the companies and their products, and also provide their own opinion about the products. However, as concluded by Grant et al. [61], such systems will

only succeed if they supplement social interactions rather than supersede them, if there is true demand for their use.

The interviews with consumers in Härnösand indicate that many of them are willing to support a resilient food system by making conscious shopping decisions, but they demand more information about where the food is produced, how it is produced, and how it affects health and the environment. Consumer habits are not static and policy instruments and educational campaigns can be used to catalyse a shift towards a more plant-based diet, reduce food waste, or promote other habits that are associated with a resilient and sustainable food system [14]. Dissemination of knowledge about consumer behaviour and choices regarding taste, price, availability, health and sustainability, social norms, and marketing is important for developing a more sustainable food consumption. Some attempts to map these habits in the region have been made but further and more systematic mapping is necessary. Reports from local TV and newspapers suggest that consumers tend to favour locally produced food during the COVID-19 pandemic. The intensified interest in food systems and changed consumer demand may be used as a lever in the transition towards a more resilient food system.

Abson et al. [13] argue that, to date, sustainability research and policy have primarily addressed relatively shallow leverage points with low transformative power to create a biophysically sustainable and socially just world. The COVID-19 pandemic has the potential to enact policy interventions and change the scientific discourse at deep leverage points that may bring about greater societal transformations [12,62]. The COVID-19 pandemic has already instigated policy changes, behavioural changes, and formation of new alliances, resulting in increased home gardening, decreased food waste, and increased procurement of local food products. COVID-19 has thus acted as a catalyst, accelerating a transition of the food system in Härnösand towards a more sustainable and resilient direction. This development can potentially stimulate the use of local/regional by-products for innovative food production purposes. Equally important is the municipality's interest in supporting sustainable food production. This, combined with current policy ambitions and more locally adapted policies, can stimulate IS symbiosis towards a more sustainable and resilient food system. A post-COVID-19 economic recovery focused on increased consumption risks to be incompatible with many sustainable development goals, but if "green" recovery measures are prioritized, the economic re-launch may offer a unique opportunity to invest in sustainable business models, including a resilient food sector based on IS [63]. Mukanjari and Sterner [64] argue that efforts to revitalize the economy should target green investments but emphasize that the identification of businesses that make sound sustainable investments is complex. It is of course too early to predict the long-term effects of the COVID-19 pandemic as a catalyst for developing a resilient food system in Härnösand, but a number of opportunities, as well as possible pitfalls, have been identified. Actors who aspire to forge a resilient food system in Härnösand need to be innovative and seek inspiration from well-established food systems worldwide. In mobilising towards productive utilisation of residues and by-products in sustainable food systems in Härnösand, theoretical frameworks such as the business model canvas for a circular economy [65], doughnut economics [21] or the framework for strategic sustainable development (FSSD) [66] may prove useful to safeguard that the development of innovative systems occur within planetary as well as social boundaries.

## 5. Conclusions

The increased awareness surrounding the importance of resilience in terms of food and water supply, health, and economy due to the COVID-19 pandemic seem to have instigated institutional adaptations in Härnösand that may translate into transformative actions towards a resilient food system in the long term. The material flow inventory revealed that many underexploited resource flows are present in quantities that render them commercially interesting. Resources that can be used for innovative food production include, e.g., lignocellulosic residues, rock dust, and food processing waste. The large share

of abandoned agricultural land suggests that agricultural production could be substantially increased without conflicting with forest production or nature conservation. A systematic mapping of available resources, categorization of their potential uses for food production, and redirection of material flows through IS symbiosis, bear the potential to stimulate innovation towards achieving a more resilient food system in Härnösand. The internalised drive among local companies interested in IS and the emerging symbiotic relations between a few of them, provide fertile ground for the establishment of a local IS network that can process the sub-exploited material flows. Long distances between facilities and the fact that the most suitable land for new developments within an IS network is not currently owned by the municipality highlight limiting factors that need to be addressed. Experiences from other regions reveal that a steering group is crucial for IS networks to materialise. The municipality of Härnösand may play a particularly important role in coordinating and conducting symbiosis development activities between different actors involved in the food system. Although there are multiple challenges for an IS network to form in Härnösand, this study shows that there is a significant potential to create added value from the region's many resources while making the food system more sustainable and resilient, by expanding IS symbiosis practices.

**Author Contributions:** Conceptualization, H.H., A.-S.F. and P.C.; methodology, H.H., A.-S.F., P.C., P.v.d.B. and O.E.; investigation, H.H., A.-S.F., P.C., P.v.d.B., O.E., W.S., I.D., K.B. and M.M.; writing—original draft preparation, H.H., A.-S.F., O.E. and P.v.d.B.; writing—review and editing, H.H., A.-S.F., P.C., P.v.d.B., O.E., W.S., I.D., K.B. and M.M.; visualization, H.H. and A.-S.F. All authors have read and agreed to the published version of the manuscript.

**Funding:** This research was funded by a cooperation agreement between Mid Sweden University and the municipality of Härnösand.

**Informed Consent Statement:** Not applicable.

**Conflicts of Interest:** The authors declare no conflict of interest.

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
