# Peer review of "Towards a Resilient and Resource-Efficient Local Food System Based on Industrial Symbiosis in Härnösand: A Swedish Case Study"

_sustainability, doi:10.3390/su14042197_

Round 1

Reviewer 1 Report

Thanks for revisions. 

Reviewer 2 Report

The efforts for increasing the value of the paper are highly appreciated.

Reviewer 3 Report

In the current form this manuscript is ready to be published

This manuscript is a resubmission of an earlier submission. The following is a list of the peer review reports and author responses from that submission.

Round 1

Reviewer 1 Report

TITLE:

Please don’t use ‘.’ at the end of title.

ABSTRACT:

The term ‘industrial symbiosis’ may explain in this part briefly. It doesn’t clear. Please elaborate it.

KEYWORDS:

Please Prefer  ‘food chain’ or ‘food supply chain’ instead of ‘food security'

INTRODUCTION

appropriate

MATERIAL-METHOD

appropriate

RESULTS:

appropriate

DISCUSSION:

appropriate

CONLUCISON:

appropriate

FUNDING:

Please type the funding number if possible.  This type is not formal.

REFERENCES:

The manuscript contains too many references for a research paper. Please don’t use general/common ones and prefer the current ones. It look likes a review

Please check and correct the typos

Author Response

Thank you for your valuable input. Please find below how you points have been incorporated.

Please don’t use ‘.’ at the end of title.

The”.” was removed

The term ‘industrial symbiosis’ may explain in this part briefly. It doesn’t clear. Please elaborate it.

Since we already have used 250 words in the abstract following the Background-Methods-Results-Conclusion-structure recommended in ”Instructions for authors” we didn’t want to burden the abstract with more text. Industrial symbiosis is an established concept and on lines 82-88 in the introduction, we describe explain the concept.

Please Prefer  ‘food chain’ or ‘food supply chain’ instead of ‘food security'

The key word ”food security” was changed to” food supply chain”

Please type the funding number if possible.  This type is not formal.

There is no funding number for this, since it is an internal fund.

The manuscript contains too many references for a research paper. Please don’t use general/common ones and prefer the current ones. It look likes a review

Although we agree that 66 is a high number of references, we believe is it justified for this kind of article. The first 28 citations are found in the introduction and methods sections were we have intended to support each key point with one to three citations. In the results, we present 1) an inventory of useful waste flows and by-products and 2) a screening literature search to identify potential applications of residues and waste products in food production. We believe that the review-like natural of these two sections justifies a higher number of references. In the following two sections were we present more original research, unique to this project, we cite considerably less references (2 citations in 3.2.1 and 9 citations in 3.3).

Reviewer 2 Report

The paper "Towards a resilient and resource-efficient local food system based on industrial symbiosis: a case study for Härnösand, Sweden" is timely and relevant to the sustainable development agenda. This is an interesting article on a topic of high interest and it is my pleasure to review it.

The paper has merits. The literature is dense and comprehensive, with well-chosen contributions, suitable for the objectives of the paper. 

I would recommend the authors to add a methodology flowchart to help the journal reader to get into the flow of the study conducted. Materials and methods used in the paper should be clearly presented in a logical format in order to help a reader to grasp the following. There seem to be a mixed up of all the element which imped the transparency and accountability of the research process itself.  There is no information about sample representativeness of the study.

Overall, this manuscript has well base of theory and concepts to examine  opportunities and challenges for using waste flows and by-products for local food production, facilitated by Industrial Symbiosis. All results are clearly and appropriately presented. 

Author Response

Thank you for your valuable input. Please find below how you points have been incorporated.

I would recommend the authors to add a methodology flowchart to help the journal reader to get into the flow of the study conducted.

A methodology flow chart was added as a new figure (figure 1)

Materials and methods used in the paper should be clearly presented in a logical format in order to help a reader to grasp the following. There seem to be a mixed up of all the element which imped the transparency and accountability of the research process itself. 

We believe that the addition of the flow chart and the rewritings and additional text about sample representativeness increases the transparency of the text and makes it easier to follow.

There is no information about sample representativeness of the study.

The chain sampling (or snowball sampling)-method that we used, admittedly has its drawbacks and advantages. One drawback is the potential lack of representativeness and community bias that may result due to the first participant’s strong impact on the sample. However, we believe that chain sampling was an appropriate technique in this context since:

1). we dealt with a limited number of stakeholders in a small municipality

2). the researchers have had extensive previous contact with the target groups.

3). the method was used as a complementary methodology together with other research methods.

We have now rewritten this section to explain the choice of method in relation to its representativeness.

Reviewer 3 Report

Dear authors, 

It was a pleasure to review your article. I find it very interesting, with high originality and an adequate methodology. The results are thoroughly presented and the paper shows a high level of professionalism.  However, to make the paper more reader-friendly I highly recommend a more graphical presentation of the results. 

Author Response

Thank you for your valuable input. Please find below how you points have been incorporated.

 However, to make the paper more reader-friendly I highly recommend a more graphical presentation of the results.

A table (table 1) was added to summarize the results from sections 3.3.1-3.3.3

Reviewer 4 Report

This is a fascinating paper on a very important topic. You (authors) provide an excellent overview of the challenges facing the shift to more sustainable food systems, particularly in the European context. Your framing of COVID-19 as a "catalyst" for change to economic model is very optimistic but given the remit of the special issue this framing is appropriate. Rather than "We also discuss how the COVID-19 pandemic may act as a catalyst for the transition towards such systems by instigating policy changes, behavioural changes and formation of new alliances" (22) I suggest that this is a major premise of your article.

Overall, the paper is much more ambitious and broad in scope than the role of Industrial Symbiosis (IS) in a sustainable food system (there are four significant objectives, can these be refined?). To narrow the focus for the reader, I recommend that you elaborate on the concepts of circular economy and of doughnut (economics) to demonstrate, explicitly, where IS fits in these models. That might also help you to develop an organising framework for the presentation of data and discussion of results, which is not easy to follow at this stage of the draft. In the Results section especially the topics jump from forestry to livestock to fisheries. Summarising the various outputs amenable to IS in the opening paragraph would make the following explanations easier to navigate for the reader. 

Food security is mentioned but opportunities are missed to highlight the fact that any just solutions to environmental problems must also account for social injustices, inequity and lack of participatory parity.

Further signposting for the reader might include a summary of enabling conditions - e.g. scale-specific policy; location, in case of Härnösand -  to facilitate the "incremental innovation approach".

Minor editorial comments include:
"one meat cutting plant" - should this be "processing"? 284

Edit for clarity - "Processes that enable the creation and diffusion of knowledge regarding multiple aspects of business opportunities linked to underexploited waste flows play a vital role for the emergence of industrial symbiosis practices." 335

"instituional" 341

Edit grammar e.g. "The awareness of the need for such a transition have 
reached a broader audience due to the pandemic."

Author Response

Thank you for your valuable input. Please find below how you points have been incorporated.

Rather than "We also discuss how the COVID-19 pandemic may act as a catalyst for the transition towards such systems by instigating policy changes, behavioural changes and formation of new alliances" (22) I suggest that this is a major premise of your article.

We agree that this is a major premise of the article and the word “also” was changed to “successively” to reflect this.

Overall, the paper is much more ambitious and broad in scope than the role of Industrial Symbiosis (IS) in a sustainable food system (there are four significant objectives, can these be refined?).

Although the scope of the paper is indeed broad, the results are delimited to the potential of Industrial Symbiosis (IS) to promote a sustainable food system. In order to emphasize that the aim of the article focuses on the role of IS in a sustainable food system (discussed in a broader context) the aim (line 141-143) was rephrased to mirror the focus on IS.

 To narrow the focus for the reader, I recommend that you elaborate on the concepts of circular economy and of doughnut (economics) to demonstrate, explicitly, where IS fits in these models.

New sentences were added to clarify the role of IS within circular economy (lines 92-92). We also elaborated the fragment on doughnut economics to bring out its potential to promote circular resource use and highlight the importance of a social foundation (line 126-128).

That might also help you to develop an organising framework for the presentation of data and discussion of results, which is not easy to follow at this stage of the draft.

Various sentences were rephrased to increase transparency and an opening paragraph in the results section was added to help the reader to navigate (246-251). A new flow chart was added in the methods section (figure 1) to describe how the methodological strategy was organised. In the results section, we added a table to present the data in a more reader-friendly way.

In the Results section especially the topics jump from forestry to livestock to fisheries. Summarising the various outputs amenable to IS in the opening paragraph would make the following explanations easier to navigate for the reader. 

An opening paragraph, summarising the various outputs amenable to IS was added for coherence (246-25).

Food security is mentioned but opportunities are missed to highlight the fact that any just solutions to environmental problems must also account for social injustices, inequity and lack of participatory parity.

This is a very good point and we have now:

  • emphasized the importance of a social foundation by adding some explanatory lines in relation to the description of doughnut economics (line 126-129)
  • added a new sentence (lines 558-580) to emphasize the importance of meeting peoples need in order to be considered sustainable
  • the final sentence in the discussion was rephrased and expanded (lines 639-644)

 Further signposting for the reader might include a summary of enabling conditions - e.g. scale-specific policy; location, in case of Härnösand -  to facilitate the "incremental innovation approach".

The table (table 1) that was added summarizes enabling conditions and limitations.

Minor editorial comments include:
"one meat cutting plant" - should this be "processing"? 284

”Meat cutting plant” was changed to "processing plant for meat”

Edit for clarity - "Processes that enable the creation and diffusion of knowledge regarding multiple aspects of business opportunities linked to underexploited waste flows play a vital role for the emergence of industrial symbiosis practices." 335

The sentence was rephrased for clarity (line 351-353)

"instituional" 341

Changed to institutional

 Edit grammar e.g. "The awareness of the need for such a transition have reached a broader audience due to the pandemic."

The sentence was rephrased (lines 113-115)